# Multivariate Analysis and Machine Learning for Ripeness Classification of Cape Gooseberry Fruits

**Miguel De-la-Torre** [1], **Omar Zatarain** [1], **Himer Avila-George** [1,*], **Mirna Muñoz** [2], **Jimy Oblitas** [3], **Russel Lozada** [4], **Jezreel Mejía** [2] **and Wilson Castro** [3,5]

1. Departamento de Ciencias Computacionales e Ingenierías, Universidad de Guadalajara, Ameca 46600, Jalisco, Mexico; miguel.dgomora@academicos.udg.mx (M.D.-l.-T.); omar.zatarain@academicos.udg.mx (O.Z.)
2. Centro de Investigación en Matemáticas, Zacatecas 98160, Mexico; mirna.munoz@cimat.mx (M.M.); jmejia@cimat.mx (J.M.)
3. Facultad de Ingeniería, Universidad Privada del Norte, Cajamarca 06001, Peru; jimy.oblitas@upn.edu.pe
4. Escuela Profesional de Ingeniería Electrónica, Facultad de Producción y Servicios, Universidad Nacional de San Agustín, Arequipa 04000, Peru; rlozadav@unsa.edu.pe
5. Facultad de Ingeniería de Industrias Alimentarias, Universidad Nacional de Frontera, Sullana 20100, Peru; wcastro@unfs.edu.pe
* Correspondence: himer.avila@academicos.udg.mx
† This paper is an extended version of our paper published in Proceedings of the 8th International Conference on Software Process Improvement, León, Guanajuato, México, 23–25 October 2019.

**Abstract:** This paper explores five multivariate techniques for information fusion on sorting the visual ripeness of Cape gooseberry fruits (principal component analysis, linear discriminant analysis, independent component analysis, eigenvector centrality feature selection, and multi-cluster feature selection.) These techniques are applied to the concatenated channels corresponding to red, green, and blue (RGB), hue, saturation, value (HSV), and lightness, red/green value, and blue/yellow value (L*a*b) color spaces (9 features in total). Machine learning techniques have been reported for sorting the Cape gooseberry fruits' ripeness. Classifiers such as neural networks, support vector machines, and nearest neighbors discriminate on fruit samples using different color spaces. Despite the color spaces being equivalent up to a transformation, a few classifiers enable better performances due to differences in the pixel distribution of samples. Experimental results show that selection and combination of color channels allow classifiers to reach similar levels of accuracy; however, combination methods still require higher computational complexity. The highest level of accuracy was obtained using the seven-dimensional principal component analysis feature space.

**Keywords:** Cape gooseberry; color space selection; color space combination; food engineering

## 1. Introduction

In the advent of the fourth industrial revolution, the growing tendency of automation of human activities encourages the use of robotic systems in the food industry [1]. In this context, the automation of food packing processes is essential to accelerating the production rate, and reducing human contact and possible contamination of food products. Moreover, machine vision techniques allow robotic systems to retrieve information from food products, using different sensors that depend on the particular characteristics to be measured, and each sensor represents an additional cost to construct an information retrieval system. For instance, an application that requires such automation systems is the classification of Cape gooseberry (*Physalis peruviana* L.) fruits according to their level of ripeness. Current industry practices address this repetitive task through visual inspection of color, size, and

shape parameters [2]. While automated sorting systems based on computer vision techniques have been proposed to improve production methods and provide high-quality products, their operation relies on classification algorithms that consider either different color spaces or a combination of them [3,4].

The most common representation of color images employed by computer vision systems is a combination of the three primary colors: Red, green, and blue (RGB). The triplet with the values for each primary color is typically considered as a coordinate system with either Euclidean, Mahalanobis, Hamming, or a different metric of distance. In such a three-dimensional coordinate system, each point (e.g., 3D vector) represents a different color in the visible spectrum. Other color spaces different than RGB are commonly employed, providing different three-dimensional representations, and can be classified into three categories according to [5]: Hardware-orientated spaces, human-orientated spaces, and instrumental spaces. In the first category, hardware-orientated spaces (e.g., RGB, YIQ, and CMYK ) are defined based on the properties of the hardware devices used to display images. On the other hand, human-orientated spaces (e.g., HSI, HSL, HSV, and HSB ) are based on hue and saturation, following the principles of an artist and based on inherent color characteristics. Finally, instrumental spaces (e.g., XYZ, L*a*b*, and L*u*v* ) are those used for color instruments, where the color coordinates of an instrumental space are the same on all output media.

As will be considered in Section 2, the color spaces that are most commonly employed in the classification of fruits are RGB, L*a*b*, and HSV. However, the accuracy of the same classifier on the same dataset may vary from one color space to the other. Some authors have investigated these differences in classification accuracy due to the variation of the distribution of pixels in distinct color spaces or the use of different segmentation techniques. According to [6,7], the practice of color measurement in food engineering, the L*a*b* color space, is the most commonly used. The main reasons are related to the uniform distribution of colors and because the L*a*b* is perceptually uniform (i.e., equal changes in data are visually perceived as equal changes in the color space). However, it is known that color spaces like RGB, L*a*b*, and HSV are equivalent up to a transformation.

Regardless of the color space used for classification, the objective of classifiers applied to fruit sorting consists of finding a criterion to separate samples from one or other ripeness levels in the so-called feature space. The goal is to establish a decision boundary that may be applied as a fixed borderline between categories. Supervised classifiers employ labeled samples to learn a model that is used to predict a category in new, never seen unlabeled samples. Supervised classifiers commonly employed in the food industry include support vector machines (SVM), k-nearest neighbors (KNN), artificial neural networks (ANN), and decision tree (DT) [8,9].

In practice, any pattern classifier may be employed, presenting a trade-off between accuracy and complexity. While the equivalence between color spaces is well-known [10], it has been found that different color spaces allow the same classifier to produce distinct classification rates, due to variations in the distribution of color samples [3,11]. Moreover, the combination of color spaces using multivariate analysis may provide a feature space where an increase in classification accuracy is possible. For instance, in [3], a methodology to compare different combinations of machine learning techniques and color spaces (RGB, HSV, and L*a*b*) was proposed in order to evaluate their ability to classify Cape gooseberry fruits. The results showed that the classification of Cape gooseberry fruits by their ripeness level was sensitive to both the color space and the classification technique used. The models based on the L*a*b* color space and the support vector machine (SVM) classifier showed the highest performance regardless of the color space. An improvement was obtained by employing principal component analysis (PCA) for the combination of the three-color spaces at the expense of increased complexity. An extension of such a study was proposed in [4], where a supervised multivariate analysis method was compared with previous results (linear discriminant analysis, LDA).

In this paper, an extension of previous work described in [3,4] is proposed to compare multivariate analysis methods and machine learning techniques for ripeness classification. The color channels from RGB, HSV, and L*a*b* color spaces were concatenated to spam a nine-dimensional feature

space. The five multivariate methods employed to combine information from the nine color channels include PCA, LDA, independent component analysis (ICA), multi-cluster feature selection (MCFS), and eigenvector centrality feature selection (ECFS). In the last case, selection methods applied to find the most relevant features for classification were MCFS and ECFS. The main contribution of this paper is the use of multivariate techniques to find an appropriate feature space for classification.

The manuscript is organized as follows. Section 2 summarizes the most recent works published on ripeness classification, including diverse approaches and methodologies. Some of the most popular methods were selected for this comparison, and Section 3 describes the material and methods employed in experiments to compare the distinct approaches. Section 4 presents the results and a discussion on the relevant findings. Finally, Section 5 presents conclusions and future work.

## 2. Ripeness Classification

As reported in the literature, different color spaces have been used to create automated fruit classification systems, presenting different levels of accuracy that are related to both, the color space and the sorting algorithm. Table 1 shows common methods and color spaces reported in the literature used to classify distinct fruits according to their ripeness level.

**Table 1.** Color spaces and classification approaches for fruit classification in literature. NA stands for non-available information MDA stands for multiple discriminant analysis, QDA for quadratic discriminant analysis, PLSR for partial least squares regression, RF for the random forest, and CNN for the convolutional neural network. The table was taken from [3] and updated with new findings.

| Item | Colorspace | Classification Method | Accuracy | Ref |
|------|-----------|----------------------|----------|-----|
| Apple | HSI | SVM | 95 | [12] |
| Apple | L*a*b* | MDA | 100 | [13] |
| Avocado | RGB | K-Means | 82.22 | [14] |
| Banana | L*a*b* | LDA | 98 | [15] |
| Banana | RGB | ANN | 96 | [16] |
| Blueberry | RGB | KNN and SK-Means | 85-98 | [17] |
| Date | RGB | K-Means | 99.6 | [18] |
| Lime | RGB | ANN | 100 | [19] |
| Mango | RGB | SVM | 96 | [5] |
| Mango | L*a*b* | MDA | 90 | [20] |
| Mango | L*a*b* | LS-SVM | 88 | [21] |
| Oil palm | L*a*b* | ANN | 91.67 | [22] |
| Pepper | HSV | SVM | 93.89 | [23] |
| Persimmon | RGB + L*a*b* | QDA | 90.24 | [24] |
| Tomato | HSV | SVM | 90.8 | [25] |
| Tomato | RGB | DT | 94.29 | [26] |
| Tomato | RGB | LDA | 81 | [27] |
| Tomato | L*a*b* | ANN | 96 | [28] |
| Watermelon | YCbCr | ANN | 86.51 | [29] |
| Soya | HSI | ANN | 95.7 | [8] |
| Banana | RGB | Fuzzy logic | NA | [9] |
| Banana | RGB | CNN | 87 | [9] |
| Watermelon | VIS/NIR | ANN | 80 | [30] |
| Watermelon | RGB | ANN | 73.33 | [31] |
| Tomato | FTIR | SVM | 99 | [32] |
| Kiwi | Chemometrics MOS E-nose | PLSR, SVM, RF | 99.4 | [33] |
| Coffee | RGB + L*a*b* + Luv + YCbCr + HSV | SVM | 92 | [34] |
| Cape Gooseberry | RGB + HSV + L*a*b* | ANN, DT, SVM and KNN | 93.02 | [3,4] |

According to Table 1, the most common color space used for classification is RGB, with 50% of the works, followed by L*a*b* with 32%, and HSV with 14%. Similarly, the most common classifiers are ANN and SVM, with 32% of the experiments reporting results using color spaces that include RGB,

L*a*b*, and HSV. The accuracy obtained by each approach depends on the experimental settings and are not comparable at this point. However, reported accuracy ranges between 73 and 100 percent.

## 2.1. Methods for Color Selection and Extraction

The distribution of samples in the feature space depends on the measurements obtained from sensors, and in this case, the color channels for the distinct color spaces. The search for the color channels that are most relevant for classification is important to help classifiers to find the decision frontier between classes. Features that are noisy or not relevant may difficult classification problems and may conduce to a low performance even by the most sophisticated classifiers. Finding a subset of the $k$ most relevant features, either by selecting them or applying feature extraction techniques, favors the reduction of redundant and irrelevant information. The so obtained $k$-dimensional feature space employed for classification instead of the original $d$-dimensional feature space is suitable to facilitate finding a separation criterion between classes. Whereas feature extraction algorithms find a mapping to a new feature space, feature selection methods aim to select a subset of vectors that spans a feature subspace that facilitates classification.

Feature extraction approaches can be categorized according to the use of data labels in supervised and unsupervised. Unsupervised feature extraction techniques consider the underlying distribution of data solely, and aim to find a mapping to a new feature space with desired characteristics. An example of unsupervised methods is PCA, which employs the Eigenvectors of the covariance matrix of samples to maximizes their spread in each new axis. Additionally, supervised approaches employ the information from class-labels to find the mapping. A representative supervised approach is the linear discriminant analysis (LDA), that aims to maximize the spread of samples distinct classes, and minimize the within-class spread.

Analogously, supervised feature selection considers class labels to find the most relevant features, and unsupervised feature selection strategies are based exclusively on the underlying distribution of samples. The selection of the subset of the most relevant features is a computational expensive combinatorial problem. The optimally of an algorithm to find a good enough feature subset may depend on the strategy followed for ranking or selection of features.

In feature selection and extraction, the problem can be stated by considering a set of points (sample tuples or feature vectors) $X = \{x_1, x_2, ..., x_N\}$, where $x_i \in R^d$. The algorithms for feature extraction and selection, find a new set $X' = \{x'_1, x'_2, ..., x'_k\}$, where $x'_i \in R^k$, and $k \leq d$ is the new dimension of the feature vectors.

## 2.2. Principal Component Analysis (PCA)

The PCA method is applied to find a linear transformation that finds the directions of maximum variance data. Sample patterns are projected onto a new feature space, and the axes with more explained variance provide a distribution that facilitates the separation between classes. The algorithm is shown in the Figure 1 depicts the general procedure to transform data samples from X to their new representation in the $k$-dimensional feature space X'. The new $k$-dimensional feature space corresponds to the $k$ eigenvectors of the covariance matrix $C$. The axis with the highest eigenvalues expresses a higher explained variance.

> **Input:** $X$ <- Set of $d$-dimensional data samples
> **Output:** Samples $X'$ mapped to the new $k$-dimensional feature space
> 1 Normalize samples in the input $d$-dimensional data
> 2 Compute covariance matrix $C_{d \times d}$
> 3 Decompose the covariance matrix $C$ into its eigenvectors
> 4 Select the $k$ eigenvectors with largest eigenvalues ($k \leq d$)
> 5 Generate a $d \times k$ projection matrix $W$ from the highest $k$ eigenvectors
> 6 Map the samples to the new $k$-dimensional feature space using $X' = X \cdot W$

**Figure 1.** The procedure followed by principal component analysis (PCA) to map the input data samples to the new $k$-dimensional feature space.

### 2.3. Linear Discriminant Analysis (LDA)

The LDA method allows obtaining and applying a linear transformation that finds the directions of maximum variance input data samples. The main difference with PCA is that LDA aims to minimize intraclass variability, whereas it maximizes interclass variability employing class labels. The main limitation is that the number of classes bounds the number of features in the new $k$-dimensional space (e.g., $1 < k < c$, where $c$ is the number of classes). This limitation makes this approach advantageous only with a high number of classes, and unpractical for data with a few classes (e.g., $c << d$). The procedure followed in computing the mapping and transforming the data is shown in Figure 2.

> **Input:** $X$ <- Set of $d$-dimensional data samples
> $\quad\quad$ $L$ <- Set of labels for the $N$ data samples in $X$
> **Output:** Samples $X'$ mapped to the new $k$-dimensional feature space
> 1 Standardize the samples in the $d$-dimensional dataset
> 2 Compute the $d$-dimensional mean vector corresponding to each class
> 3 Construct the between-class scatter matrix $S_B$, and the within-class scatter matrix $S_w$
> 4 Compute the Eigenvectors and corresponding Eigenvalues of the matrix $S_w^{-1} S_B$
> 5 Generate a $(d \times k)$-dimensional transformation matrix $W_{LDA}$ with the $k$ eigenvectors with the $k$ largest eigenvalues as columns
> 6 Map the samples onto the new feature subspace using $W_{LDA}$

**Figure 2.** The procedure followed by linear discriminant analysis (LDA) to map the data samples to the new $k$-dimensional feature space.

### 2.4. Independent Component Analysis (ICA)

The ICA method finds underlying components from multivariate statistical data, where data is decomposed into components that are maximally independent in an appropriate sense (e.g., kurtosis and negentropy). The difference between PCA and LDA is that low-dimensional signals do not necessarily correspond to the directions of maximum variance; rather, the ICA components have maximal statistical independence and are nongaussian. In practice, ICA can be used to find disjoint underlying trends in multi-dimensional data [35].

The algorithm is shown in the Figure 3 depicts the procedure followed by the FastICA algorithm to obtain the independent components from $X$, using kurtosis as a measure of non-gaussianity. In this case, dimensionality reduction is not obvious, given that there is no measure of how important a particular independent component is. The relevance of the individual features obtained with PCA and LDA is given by the algorithms shown in Figures 1 and 2, respectively. In the case of ICA, feature selection techniques may be employed to provide a relevance level for each of the features that result from the transformation, as described in Sections 2.5 and 2.6.

> **Input:** $X$ <- Set of $d$-dimensional data samples
> **Output:** Samples $X'$ mapped to the new $k$-dimensional feature space $(k = d)$
> 1 Center data samples in $X_c = center(X)$
> 2 Whiten data and obtain $X_w = D^{1/2}E^T X_c$, where $E$ is the matrix of Eigenvectors, and $D$ is the diagonal matrix of Eigenvalues of the covariance matrix of $X_c$
> 3 Set a random initial weight $N$-dimensional vectors $\mathbf{w} = rand(N)$
> 4 Estimate nongaussianity according to kurtosis; $G = 4s$, and $Gp = 12S_k^2$
> 5 Update weight $\mathbf{w} = GX_{pw} - Gp\mathbf{w}$
> 6 If $\Delta = max(1 - |W, W_{prev}) > \gamma$, for a given threshold $\gamma$, go back to line 4
> 7 Repeat for each vector $\mathbf{w}$ to produce the new mapping $W$
> 8 $X' = WX$ is the new representation of data in the new feature space

**Figure 3.** The procedure followed by FastICA to map data samples in $X$ to the new feature space that respects nongaussianity using kurtosis.

### 2.5. Eigenvector Centrality Feature Selection (ECFS)

The feature selection via eigenvector centrality is a supervised method that ranks features by identifying the most important ones. It maps the selection problem to an affinity graph with features as nodes and assesses the rank features according to the eigenvector centrality (EC) [36].

The algorithm shown in the Figure 4 presents the method to rank and select the most relevant features from the data samples $X$. While this does not constitute a proper transformation in terms of linear algebra, every sample is represented in a new $k$-dimensional feature space with the highest-ranked features.

> **Input:** $X$ <- Set of $d$-dimensional data samples
> $L$ <- Set of labels for the $N$ data samples in $X$
> **Output:** $v_0$ ranking scores for each of the $d$ features
> 1 Build the weighted graph $G = (V, E)$; where vertices $V$ correspond to data samples and edges $E$ among features:
> 2 Compute the Fisher score : $f_i = \frac{\sum_{c=1}^C (\mu_{i,C} - \mu_i)^2}{\sigma_i^2}$ , where $\mu_{ij}$ represents the mean, and $\sigma_{ij}$ is the standard deviation for the whole dataset corresponding to feature $i$
> 3 Compute mutual information: $m_i = \sum_{y \in Y} \sum_{z \in x^{(i)}} p(z, y) \log \left( \frac{p(z,y)}{p(z)p(y)} \right)$, where $p(\cdot, \cdot)$ is the joint probability distribution
> Obtain the adjacency matrix $A = \alpha k + (1 - \alpha)\Sigma$, where $k = f \cdot m^T$ and $\Sigma(i, j) = max(\sigma^{(i)}, \sigma^{(j)})$ Ranking: Compute the eigenvalues $\{\Lambda\}$ and eigenvectors $\{V\}$ of $A$, where $\lambda_0 = max_{\lambda \in \Lambda}(|\lambda|)$
> 5 $v_0$ is the eigenvector associated to $\lambda_0$

**Figure 4.** The procedure followed by eigenvector centrality feature selection to select the variables that constitute the new feature space.

### 2.6. Multi Cluster Feature Selection (MCFS)

Multi-class feature selection (MCFS) is an unsupervised technique that aims to find those features that preserve the multi-cluster underlying structure of the samples used for training [37]. Given that the number of clusters is unknown a priori, it is a good practice to explore distinct values to find a good feature subspace. The most relevant features are found following the procedure shown in the algorithm shown in the Figure 5.

> **Input:** $X$ <- Set of $N$ $d$-dimensional data samples
>   $K$ <- Number of clusters, default $K = 5$
> **Output:** The top $k$ features according to their MCFS scores
> 1 Construct a $p$-nearest neighbor graph Laplacian (e.g., $W_{ij} = 1$ `iif` $i$ and $j$ are connected by an edge;
>   $D_{ii} = \sum_j W_{ij}$ ; and $L = D - W$)
> 2 Solve the generalized eigenproblem $Ly = \lambda Dy$, where $Y = y_1, ..., y_K$ are the top $K$ eigenvectors with
>   respect to the smallest eigenvalues.
> 3 Solve the regression problem: $\min_{a_k} ||y_k - X^T a_k||^2$, with a user-defined cardinality to control the
>   sparseness of $a_k$
> 4 Compute: $MCFS(j) = \max_k |a_{k,j}|$

**Figure 5.** The procedure followed by eigenvector centrality feature selection to select the variables that constitute the new feature space.

While the simplest method to choose W was presented in Step 1, other methods exist that range between accuracy and complexity (See [37]). According to the authors, the default number of nearest neighbors is $p = 5$, and the default number of eigenfunctions is $K = 5$. This last parameter $K$ usually influences the accuracy of the algorithm and should be optimized before usage.

*2.7. Classification for Fruit Sorting*

According to Table 1, some of the most popular supervised classifiers in fruit sorting are the artificial neural networks (ANN), decision trees (DT), support vector machines (SVM), and k-nearest neighbor (KNN). These classifiers were used in this paper for the experimental settings. While these techniques have been present in the literature for many years now, see [3,38], their usage in practice increased due to their capacity to address diverse real-world problems.

ANN is a non-linear classifier that simulates biological neural networks. A common implementation of ANN corresponds to the probabilistic ANN, which produces an estimated posterior probability for each input sample to belong to any of the classes, and the max function allows to select the most likely class. In this research, the Matlab's Neural Network Toolbox was used to implement the probabilistic ANN classifier, byways of the newpnn function, tunned to optimize hyperparameters using linear search.

DT is a tree-based example of the knowledge used to represent the classification rules. Internal nodes are representations of tests of an attribute; each branch represents the outcome of the test, and leaf nodes represent class labels. In this paper, the Matlab's Machine Learning Toolbox (MLT) was used the train and simulate DTs, using the Classification & Regression Trees (CART) algorithm to create decision trees, with the fitctree and predict functions. The function fitctree employes standard classification and regression trees algorithm to create DTs.

SVM is a non-parametric statistical learning classifier that constructs a separating hyperplane (or a set of hyperplanes) in a high-dimensional feature space. Some versions use the so-called kernel trick to map data to higher dimensional feature space and find the separating hyperplane there. The functions fitcecoc and predict functions were used for simulations, both implemented in Matlab's MLT. The fitcecoc function was tunned to use a linear kernel and Bayesian hyperparameter optimization.

KNN is a non-parametric classifier that keeps all training samples, and prediction is based on the number of closest neighbors belonging to a class. Given an input sample, the distance to all stored samples is computed and compared to all pre-stored samples, presenting a high computational complexity at prediction. For simulations, the fitcknn and predict functions from Matlab's MLT were used. This function employs Bayesian hyperparameter optimization.

## 3. Materials and Methods

For experiments, a set of 925 samples of Cape gooseberry fruits were collected from a plantation located in the village of El Faro, Celendin Province, Cajamarca, Peru (UTM: $-6.906469$, $-78.257071$). Fruit samples were homogeneously disposed on a conveyor belt used in a production line ($160 \times 25$ cm,

and 80 cm high). A Halion-HA-411 VGA webcam was used for image capture, which provides RGB raw images in JPG format. The resolution of the resulting images is 1280 × 1720 pixels. The camera was fixed 35 cm over the conveyor belt, and the captured scene was covered with black matte panels to reduce variations in light, as implemented by Pedreschi et al. [39]. The illumination system included two long fluorescent tubes (Philips TL-D Super, cold daylight, 80 cm, 36 W) that were placed on both sides of the conveyor belt. Additionally, a circular fluorescent tube (Philips GX23 PH-T9, cold daylight, 21.6 cm, 22 W) was located at the top of the setting. Images captured with the camera were stored on a portable computer running Matlab to control image acquisition and data analysis.

　　Seven levels of ripeness were employed for visual classification, following the standard for Cape gooseberry, and the visual scale proposed in [40] and shown in Figure 6. The process followed for evaluation is depicted in Figure 7. Images captured from the conveyor belt (step 1) were employed to find the regions of interest corresponding to Cape gooseberry fruits in the image, employing standard segmentation techniques (steps 2 and 3); the size of resulting regions depends on the size of the actual fruit. Color versions of segmented fruits were labeled by five experts according to the categorization proposed by Fischer et al. in [40] (step 4). One-color sample was selected for each fruit region in each of the RGB, HSV and L*a*b* color spaces, by computing the average for each color channel; and a nine-dimensional feature vector was built through concatenation: $x = [R, G, B, H, S, V, L*, a*, b*]^T$ (step 5). Then, multivariate analysis techniques for feature extraction/selection were applied to the set of feature vectors (step 6), and the resulting samples were organized for five-fold cross-validation. Four standard classifiers were trained (step 7) and performance evaluation computed (step 8).

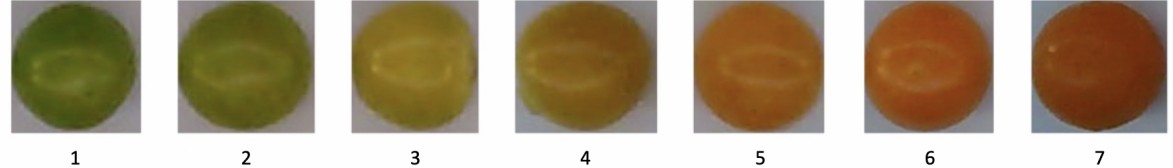

|  1  |  2  |  3  |  4  |  5  |  6  |  7  |

**Figure 6.** Levels of ripeness employed for supervised visual classification.

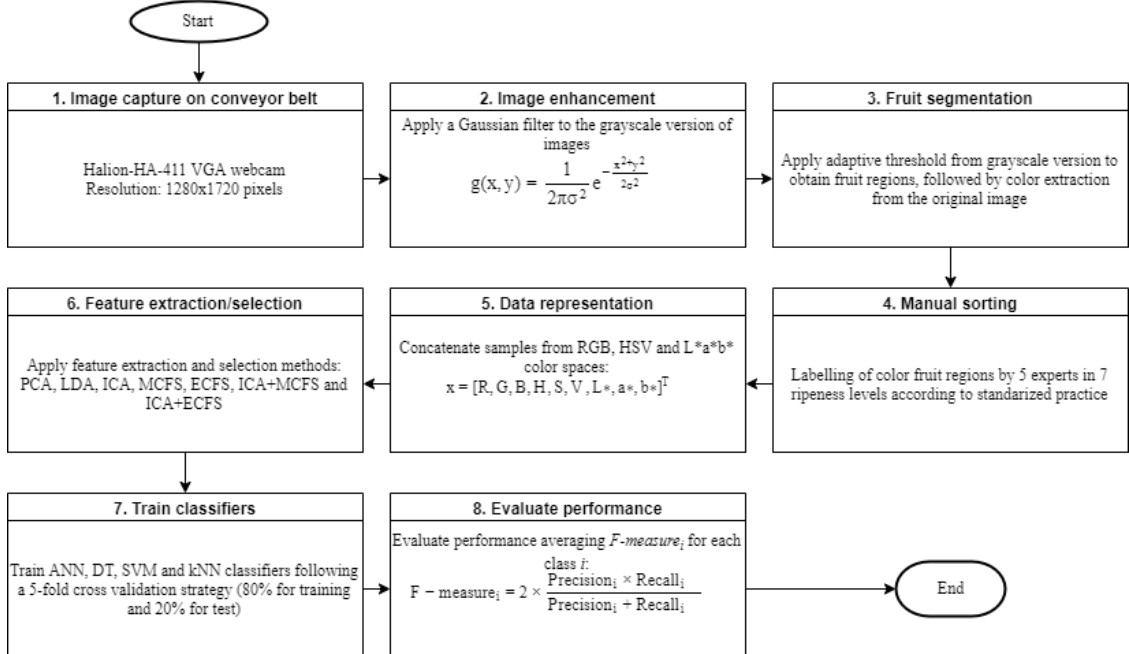

**Figure 7.** Experimental process followed to evaluate the system with distinct feature extraction/selection methods and different classifiers.

The performance of the seven-class classifiers was evaluated using the F-measure, as defined in [3]. First, the confusion matrix is computed according to the responses of each classifier, and true positives ($TP_i$), false positives ($FP_i$), true negatives ($TN_i$), and false negatives ($FN_i$) are obtained for each class $i$, using the elements $N_{ij}$ of the confusion matrix. Class-specific precision and recall are computed using Equations (1) and (2), respectively.

$$Precision_i = \frac{TP_i}{TP_i + FP_i} \tag{1}$$

$$Recall_i = \frac{TP_i}{TP_i + FN_i} \tag{2}$$

Finally, the multiclass F-measure was used for comparison along with the experimental results, due to its representativeness of the classification performance on target classes (Equation (3)).

$$F - measure_i = 2 \times \frac{Precision_i \times Recall_i}{Precision_i + Recall_i} \tag{3}$$

The three analyses followed to characterize the performance of the system started by fixing the classifier (e.g., SVM). First, the $k$ (number of clusters) was explored in order to find the $k$ that allows the highest classification performance. Then, the size of the feature space was explored in terms of average F-measure. The last analysis explores the performance using the parameters found in previous steps, and the four classifiers presented in Section 2 ANN, SVM, DT, and KNN.

## 4. Experimental Results

As explained in Section 2.6, MCFS needs a search to find the number of clusters that maximizes the classification performance. The number of characteristics was set to seven, to make it comparable with previous results using PCA [3].

Figure 8 shows the box plots that summarize the distribution of performance for the SVM classifier trained with seven color channels (features) selected with the MCFS algorithm. The parameter that controlled the number of clusters was moved from 1 to 9 (i.e., the maximum number of possible features). In most cases, the median of the F-measure was maintained around 71.75, and only two cases were different: 2 and 9. Using nine clusters appears to provide lower performance related to the creation of an excessive number of clusters. On the other hand, using only two clusters for feature selection seems to provide a higher level of accuracy. However, and regardless of the median accuracy, the variability between cases shows a difference that makes no significant difference in using a different number of clusters. Therefore, in the following experiments, the number of clusters is fixed at 2, and it explored other variables.

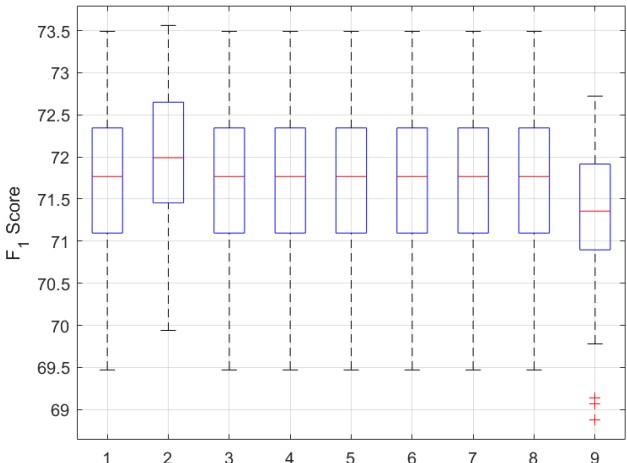

**Figure 8.** Boxplots corresponding to the F-measure for nine distinct values of the parameter controlling the number of clusters in multi-cluster feature selection (MCFS). The number of characteristics was fixed at 7, and the experimentation follows a five-fold cross-validation strategy.

### 4.1. Analysis of Feature Spaces

Table 2 shows the average F-measure and standard deviation corresponding to the outputs generated by the SVM classifier after training on feature spaces selected or extracted with the different methods explained in Section 2. The feature space was varied from $d \in \{1...9\}$ features, generating nine $d$-dimensional feature spaces for classification. The performance was estimated using a five-fold cross-validation strategy to obtain a measure of dispersion.

**Table 2.** Average F-measure of the support vector machine (SVM) classifier applied to distinct feature spaces obtained with the four methods for feature extraction/selection. IC stands for the independent component, bold numbers symbolize the highest F-measure obtained for each method, and numbers in parenthesis symbolize standard deviation.

| Method | 1D | 2D | 3D | 4D | 5D | 6D | 7D | 8D | 9D |
|---|---|---|---|---|---|---|---|---|---|
| PCA | 40.89 | 68.56 | 69.48 | 71.23 | 71.83 | 71.69 | 71.99 | 71.70 | 71.65 |
| | (0.34) | (0.91) | (0.95) | (0.82) | (0.92) | (0.70) | (0.81) | (0.92) | (0.91) |
| LDA | 52.43 | 69.10 | 69.48 | 70.05 | 70.02 | 71.48 | - | - | - |
| | (0.81) | (1.24) | (1.17) | (1.00) | (1.05) | (0.74) | - | - | - |
| ICA | 8.12 | 25.21 | 53.89 | 58.93 | 62.18 | 63.74 | 68.10 | 70.38 | 71.67 |
| | (0.40) | (0.45) | (1.16) | (1.12) | (1.16) | (1.02) | (0.91) | (0.87) | (0.90) |
| MCFS − 2 clusters | 64.74 | 65.67 | 70.04 | 70.72 | 71.02 | 71.92 | 71.99 | 71.83 | 71.66 |
| | (0.70) | (0.68) | (1.13) | (1.04) | (0.96) | (0.76) | (0.79) | (0.89) | (0.87) |
| Color channel | L*(7) | V(6) | H(4) | b*(9) | R(1) | G(2) | B(3) | S(5) | a*(8) |
| ECFS | 40.93 | 68.81 | 69.55 | 71.33 | 71.89 | 71.76 | 71.86 | 71.84 | 71.66 |
| | (0.32) | (1.18) | (1.2) | (0.72) | (0.72) | (0.79) | (0.83) | (0.79) | (0.87) |
| Color channel | G(2) | R(1) | a*(9) | b*(8) | H(4) | L*(7) | S(5) | V(6) | B(3) |
| ICA + ECFS | 23.21 | 25.18 | 28.82 | 36.14 | 51.84 | 51.70 | 61.22 | 62.71 | 71.67 |
| | (0.41) | (0.46) | (0.54) | (0.71) | (0.79) | (0.74) | (0.73) | (0.75) | (0.90) |
| IC | 2 | 1 | 9 | 8 | 4 | 7 | 5 | 6 | 3 |
| ICA + MCFS | 26.61 | 44.68 | 57.24 | 61.13 | 61.81 | 63.07 | 65.14 | 68.57 | 71.67 |
| | (0.57) | (1.07) | (1.08) | (1.09) | (1.15) | (1.26) | (1.14) | (0.84) | (0.90) |
| IC | 3 | 2 | 4 | 8 | 9 | 1 | 6 | 5 | 7 |

Results showed in Table 2 evidence that was using seven features provide a level of performance that is similar either using MCFS or PCA. On the other hand, ECFS and LDA present the highest level

of performance using five and six features, respectively, with a slightly lower average performance compared to PCA and MCFS. Moreover, in all cases, using more than three features allows classifiers to obtain a significantly higher performance with a lower standard deviation. In that sense, when a feature space with more color channels—or features—is employed, the SVM classifier presents a higher and more stable classification performance, at the expense of the evident increase in computational complexity. This is evident either if features are selected (e.g., MCFS, ECFS) or extracted (e.g., PCA, LDA). Different behavior is presented when ICA is employed for feature extraction, due to the strategy to find the independent components instead of the axis of maximum spread.

In the hypothetic case that only three-color channels were allowed, and these channels could be arbitrarily chosen from the nine provided by our three-color spaces, in this case, a selection method should be used. Then, a quick look at the 3D column of Table 2 evidences that the MCFS provided a better channel selection, achieving the highest performance level with a feature space composed of channels $[L*, V, H]$.

### 4.2. Performance across Classifiers

The comparative of performance in terms of F-measure, between the ANN, DT, SVM, and KNN classifiers, evaluated on the best d-dimensional feature space found in Section 4.1, is presented in Figure 9. The distinct feature spaces provided a different optimal number of characteristics, and those features were employed in each case. In particular, seven features were selected for PCA and MCFS, six features for LDA, and five features for ECFS. In the case of ICA, all nine features were employed to obtain the highest level of performance.

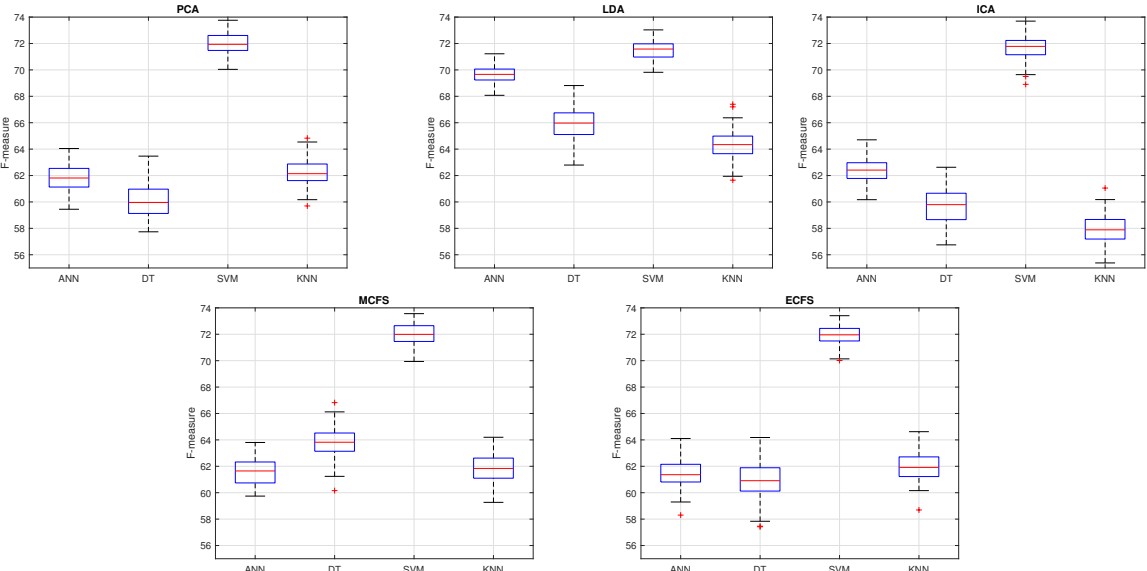

**Figure 9.** Boxplots representing the distribution of F-measure performance for the feature selection/extraction approaches, using four different classifiers.

As shown in Figure 9, the SVM classifier F-measure outperforms the rest of the approaches, and only ANN performance is close to SVM performance on the six-dimensional LDA feature space. The highest level of F-measure achieved by ANN is shown in the space extracted with LDA. In general, in these settings, the performance of all classifiers presents its highest level on the seven-dimensional PCA space. The settings suggest that PCA provides a feature space that facilitates the work of a classifier after combining information from multiple color spaces. On the other hand, focusing on the two feature selection methods, a similar level of performance is provided by all classifiers, without significant difference. The only case where KNN presents a lower performance is on the nine-dimensional ICA feature space. However, if a minimum number of features is required for a

given application (e.g., to reduce computational complexity and cost), a feature selection method may provide the means to select a few color channels (sensors), at expenses of a reduction in performance.

## 5. Conclusions and Future Work

In this paper, an extension of a food packaging process was proposed for Cape gooseberry fruit sorting, according to its ripeness level. As a difference from previous works, five techniques from the multivariate analysis were employed to find the feature spaces that facilitate classification. Whereas PCA, LDA, and ICA provided mapping to a new feature space, the two selection methods (MCFS and ECFS) provided the most relevant features for classification. The configuration of the experiments provided a realistic scenario, including accommodating Cape gooseberry fruits with distinct levels of ripeness on a conveyor belt, that were captured with a VGA camera located on top. Segmentation and manual sorting were performed before feature extraction/selection and classification. Four classifiers commonly employed in literature for ripeness classification were compared, including ANN, SVM, DT, and kNN.

Results reveal that selection and extraction methods allow classifiers to reach similar levels of accuracy, but feature extraction methods require an increased computational complexity. This evidence must be considered in a final implementation, and the real-time performance of the whole system should be observed once running with the distinct algorithms on the selected computational platform.

Considering the classifiers, the SVM classifier outperformed the rest in terms of F-measure regardless of the feature space. This may respond to the organization of samples in the feature space, and the capacity of the Bayesian optimization on SVM to find a good separating hyperplane. Moreover, the four classifiers employed in the test presented the highest level of accuracy on the seven-dimensional PCA feature space. This combination of 7-D PCA feature space and the SVM classifier should be considered when a final implementation. However, the hyperparameters for this (and other classifiers) were fixed before training, and some optimization may allow finding a higher level of performance for the distinct classifiers used in experiments.

On the other hand, the lowest level of accuracy was achieved on the one-dimensional feature space, employing the ICA feature extraction technique without a feature selection method. This evidences the need for a feature selection method when ICA is employed for finding new feature spaces with independent spanning vectors. On the opposite, the highest level of accuracy for a one-dimensional feature space was obtained with the MCFS channel selection, obtaining an F-measure of 64.74 (0.07), with a single feature (L*) using the SVM classifier. This result suggests that the L* color channel from the L*a*b* perceptually uniform color space is the feature that provides the highest degree of separation between classes. The L* feature, combined with the other six features (R, G, B, b*, H, and V), allows it to obtain a performance that is similar to PCA with the same seven features.

The results obtained in the experiments suggest some paths for further research. Future work may include the use of distinct and more sophisticated algorithms for feature selection and extraction that may be explored and combined to find the best combination for a particular application. Similarly, other algorithms for classification may be tested in this configuration, such as those employing deep learning and large data sets. Additionally, optimization techniques like particle swarm optimization or evolutionary algorithms may be employed to find the best hyperparameters for the application.

On the other hand, information fusion techniques like classifier combination strategies may also enhance the establishment of the decision borderlines between classes, with the inherent performance increase. Finally, another kind of problem may benefit from feature extraction and selection techniques in food engineering, like using multi- or hyperspectral sensors to measure the level of ripeness of Cape gooseberry or any different type of fruit.

**Author Contributions:** Individual contributions for the authors are as follows: conceptualization, W.C., H.A.-G. and M.D.-l.-T.; methodology, J.O., M.M. and J.M.; software, M.D.-l.-T. and J.O.; validation, W.C. and H.A.-G.; formal analysis, M.D.-l.-T. and O.Z.; resources, R.L., M.M. and J.M.; data curation, H.A.-G. and M.D.-l.-T.;

writing–original draft preparation, M.D.-l.-T.; writing–review and editing, O.Z., H.A.-G. and W.C.; visualization, M.D.-l.-T. and H.A.-G.; supervision, M.M. and J.M.; project administration, H.A.-G.; funding acquisition, W.C.

**Funding:** This research received no external funding.

**Conflicts of Interest:** The authors declare no conflicts of interest.

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
