# Peer review of "Multivariate Analysis and Machine Learning for Ripeness Classification of Cape Gooseberry Fruits"

_processes, doi:10.3390/pr7120928_

Round 1

Reviewer 1 Report

The authors have fused the features from different color spaces, and perform feature selection to classify the cape gooseberry fruits to their respective classes. The study sounds interesting to read would benefit the research community. However, this reviewer has a few queries:

What was the raw image format? What is the loss incurred in converting the raw image to other color spaces?

What was the original image dimension? What was the size of the resized images? What is the difference in the performance of the classifier and feature selection methods with increasing/decreasing spatial resolution?

Figure 2 is confusing. The authors mentioned having applied a Gaussian filter to the grayscale version of the images? Why did they convert the color images to grayscale? What was the color space of the original images?

The authors have mentioned to have segmented and manually annotated the original images. What is the difference in the performance of the feature selection and classification algorithms with under/over segmented data? What are the segmentation accuracy and IOU metrics computed between the ground truth and segmented images?

Did the authors normalize the images in their respective color spaces? What is the difference in the performance of feature selection/classification between mean normalized and standardized images and the non-normalized/non-standardized counterparts?

Are the results proposed generalizable to any kind of fruit classification task?

Author Response

The authors have fused the features from different color spaces, and perform feature selection to classify the cape gooseberry fruits to their respective classes. The study sounds interesting to read would benefit the research community.

Response:

The authors appreciate your useful suggestions to improve the quality of the paper. 

What was the raw image format? What is the loss incurred in converting the raw image to other color spaces?

Response:

The raw format of images given by the camera is JPG format, as provided by the Halion-HA-411 VGA webcam. This information was added to the methodology section. 

Likewise, although precision artifacts may appear due to color space conversion, the authors assume that there is minimal or no loss during the change of color space. This is mainly supported in the works of Leo et al. (2006) and Saldaña (2014); in both the equations were specifically prepared for JPG images.

• León, K., Mery, D., Pedreschi, F., & León, J. (2006). Color measurement in L*a*b* units from RGB digital images. Food Research International, 39(10), 1084e1091. http://dx.doi.org/10.1016/j.foodres.2006.03.006 • Saldana, E., Siche, R., Castro, W., Huamán, R., & Quevedo, R. (2014). Measurement parameter of color on yacon (Smallanthus sonchifolius) slices using a computer vision system. LWT-Food Science and Technology, 59(2), 1220-1226.

What was the original image dimension? What was the size of the resized images?What is the difference in the performance of the classifier and feature selection methods with increasing/decreasing spatial resolution?

Response:

The original dimension of the whole image capture is 12801720 pixels, and segmentation is performed on the whole image (never resized). The size of the resulting fruit regions varies according to the size of the actual fruit, but only the average pixel was considered to build the feature vector. This was clarified in the methodology section of the paper.

Figure 2 is confusing. The authors mentioned having applied a Gaussian filter to the grayscale version of the images? Why did they convert the color images to grayscale? What was the color space of the original images?

Response:

Figure 2 was reorganized and improved with a more clear notation; an explanation was also added with details on each step. The application of a Gaussian filter to the grayscale version of the RGB images was motivated by the standard binarization procedure employed for fruit segmentation. After segmentation, original RGB images, as well as HSV and L*a*b* variants, were stacked to form the 9-dimensional feature vector. Details on this procedure were added in the description of Figure 2.

The authors have mentioned to have segmented and manually annotated the original images. What is the difference in the performance of the feature selection and classification algorithms with under/over segmented data? What are the segmentation accuracy and IOU metrics computed between the ground truth and segmented images?

Response:

The authors manually classified the segmented images by five trained operators; thus, in the next steps, the different algorithms are compared with the manually obtained, taking this as the reference. Since images were captured in an environment with controlled illumination, there weren’t under/over segmentation cases. The following image illustrates the segmentation process: (A) original image, (B) grayscale image, (C) region of interest selection, and (D) segmented image.

(see image in the new version)

Reviewer 2 Report

This manuscript proposed a novel hybrid method for classifying Cape gooseberry fruits, which is based on multivariate analysis and machine learning techniques. In the proposed method, principal component analysis, linear discriminant analysis and independent component analysis were used for information fusion. Then, four classifiers were built up for a comparison based on captured features. Experimental results demonstrate high performance of the proposed method. Overall, the topic of this study is interesting and the manuscript is well written. It can be suggested to be published in Processes if the authors can well address the following comments.

Please illustrate the main innovation of this work in the introduction, and explain how the multivariate analysis can improve the classification accuracy. Why not using deep learning methods? In this study, the authors used four ML methods for a comparison. Please give the parameter setting of these classifiers. Or provide the details of how to get optimal parameters (hyperparameters) of classifiers. From the results in Figure 7, it is interesting that SVM outperforms the KNN classifier. Please explain this result. In conclusion, the future work should consider how to apply the proposed technique in the real application.

Author Response

Comments: This manuscript proposed a novel hybrid method for classifying Cape gooseberry fruits, which is based on multivariate analysis and machine learning techniques. In the proposed method, principal component analysis, linear discriminant analysis, and independent component analysis were used for information fusion. Then, four classifiers were built up for a comparison based on captured features. Experimental results demonstrate a high performance of the proposed method. Overall, the topic of this study is interesting, and the manuscript is well written.

Response:

Authors appreciate the comments and suggestions of the reviewer. 

Comments:Please illustrate the main innovation of this work in the introduction, and explain how the multivariate analysis can improve the classification accuracy. 

Response:

Thanks for your comment, the introduction was modified to illustrate the contribution of the paper.

Comments: Why not using deep learning methods?

Response:

Up to now, standard classification techniques were employed, providing good enough performance. Deep learning techniques are visualized to be employed in future work. This was clarified in the conclusions and future work section.

Comments: In this study, the authors used four ML methods for comparison. Please give the parameter setting of these classifiers. Or provide the details of how to get optimal parameters (hyperparameters) of classifiers.

Response:

Details on hyperparameter optimization were added for each classifier in Section 2.7. 

Comments: From the results in Figure 7, it is interesting that SVM outperforms the KNN classifier. Please explain this result.

Response:

Figure 4 reveals a significant outperformance of the SVM classifier over others. This may be due to both, the organization of samples in the feature space, and the capacity of the Bayesian optimization on SVM to find a good separating hyperplane. However, further optimization algorithms may be employed to explore the optimization space.

Comments: In conclusion, future work should consider how to apply the proposed technique in real application.

Response:

The Cape gooseberry fruit is an attractive exportation product in Perú, and it has enhanced the development of strong industrial areas surround it. However, the problem of determining the ripeness level is still a bottleneck to improve productivity and reduce production costs. Therefore, these techniques and methods have an immediate application in this and related industries. The conclusion and future work section of the paper were complemented with related information.
